# Comparative Analysis of Differentially Expressed Long Non-Coding RNA in Pre- and Postmenopausal Fibroids

**DOI:** 10.3390/ijms26146798

**Published:** 2025-07-16

**Authors:** Tsai-Der Chuang, Shawn Rysling, Nhu Ton, Daniel Baghdasarian, Omid Khorram

**Affiliations:** 1The Lundquist Institute for Biomedical Innovation, Torrance, CA 90502, USA; tchuang@lundquist.org (T.-D.C.); shawn.rysling@lundquist.org (S.R.); nhu.ton@lundquist.org (N.T.); 2Department of Obstetrics and Gynecology, Harbor-UCLA Medical Center, Torrance, CA 90509, USA; dbaghdasarian@dhs.lacounty.gov; 3Department of Obstetrics and Gynecology, David Geffen School of Medicine, University of California, Los Angeles, CA 90095, USA

**Keywords:** postmenopausal, fibroid, LncRNA, MED12, steroids

## Abstract

Uterine fibroids (leiomyomas) are benign tumors whose growth is influenced by estrogen and progesterone. This study aimed to compare the profiles of differentially expressed long non-coding RNAs (lncRNAs) in fibroids from postmenopausal and premenopausal women to identify hormone-responsive lncRNAs. RNA sequencing was performed on six pairs of fibroid (Fib) and adjacent myometrium (Myo) tissues from postmenopausal women. Out of 7876 normalized lncRNAs, 3684 were differentially expressed (≥1.5-fold), with 1702 upregulated and 1982 downregulated in Fib. Comparative analysis with a previously published premenopausal dataset identified 741 lncRNAs that were altered based on their menopausal status, including 62 lncRNAs that were uniquely dysregulated in postmenopausal samples. Overall, 9 lncRNAs were selected for validation by PCR in an expanded cohort of 31 postmenopausal and 84 premenopausal paired samples. Several lncRNAs, including *LINC02433*, *LINC01449*, *SNHG12*, *H19*, and *HOTTIP*, were upregulated in premenopausal Fib but not in postmenopausal ones, while *ZEB2-AS1* displayed the opposite pattern. *CASC15* and *MIAT* were elevated in Fib from both groups, although the increase was less pronounced in the postmenopausal group. *LINC01117* was significantly downregulated in postmenopausal Fib, with no change observed in premenopausal samples. Additionally, analysis based on MED12 mutation status revealed that lncRNAs such as *LINC01449*, *CASC15*, and *MIAT* showed limited or reduced differential expression (mutation-positive vs. mutation-negative) in postmenopausal patients compared to the premenopausal group. These findings indicate that lncRNA expression in fibroids is modulated by menopausal status, likely reflecting hormonal influence. Hormone-responsive lncRNAs may play key roles in fibroid pathogenesis and represent potential targets for therapeutic intervention.

## 1. Introduction

Fibroids (leiomyomas; Fib) are benign fibrotic tumors afflicting over 70% of reproductive-age women [1]. These tumors cause significant symptoms, including abnormal uterine bleeding, pain, and infertility, and are the main indication for hysterectomies performed [2]. The focus of our group has been to identify the pathogenic mechanism underlying the pathogenesis of fibroids, with an emphasis on the role of non-coding RNAs (ncRNA) in the regulation of protein coding genes. In a previous publication, we reported on the expression of long non-coding RNAs (lncRNA) in fibroids from premenopausal women and the influence of race/ethnicity and MED12 mutation on their expression [3]. The objective of the current study was to expand on our earlier report [3] examining the differential expression of lncRNAs in fibroids from postmenopausal women with the aim of identifying lncRNAs whose expression may be regulated by sex steroids. Both estrogen [4] and progesterone [5,6] play a pivotal role in stimulating the growth of fibroids and currently available medical therapies for fibroids all aim to reduce the production of sex hormones to effectively shrink tumor size and reduce their growth [7].

LncRNAs are single-stranded RNA molecules which are >200 nt in length, transcribed from different genomic loci [8]. These ncRNAs modulate the expression of protein coding genes through transcriptional, post-transcriptional and epigenetic mechanisms [9]. LncRNAs can function as molecular sponges of miRNA, can modify chromatin structure, act as decoys of transcription factors and regulate mRNA stability and protein function [10]. Our group and others have previously reported on the role of a number of lncRNAs in regulating key pathogenic pathways in fibroids including *XIST* [11,12], *MIAT* [13,14], *H19* [15,16], *LINCMD1* [17]. Previous studies have shown that the expression of certain lncRNAs are influenced by sex steroids [18,19]. This study provides an overall expression profile of lncRNA in fibroids from postmenopausal women with a comparison to lncRNAs in fibroids from premenopausal women thereby identifying lncRNAs whose expression could be under sex steroid control.

## 2. Results

### 2.1. High-Throughput Sequencing Analysis of lncRNA Expression in Pre- and Postmenopausal Group

To characterize the differential expression profiles of lncRNAs in postmenopausal Fib and their paired myometrium (Myo), we performed next generation sequencing using RNA isolated from six paired samples collected from postmenopausal women. After normalizing 7876 lncRNA transcripts, the hierarchical clustering and TreeView analysis revealed that 3684 lncRNAs exhibited significant differential expression in Fib compared to Myo with increased expression of 1702 lncRNAs and decreased expression of 1982 lncRNAs by at least 1.5-fold (Figure 1A). Additional volcano plot analysis (fold change ≥1.5, *p* < 0.05; Figure 1B) and principal component analysis (PCA), followed by k-means clustering, confirmed consistent expression patterns between groups indicating the reliability of our data (Figure 1C).

Next, we compared the differential expression of lncRNAs identified in this postmenopausal dataset with our previously published premenopausal dataset (GSE224991). This comparative analysis, based on the fold change in expression between Fib and paired Myo, identified 1200 differentially expressed lncRNAs, of which 741 showed altered expression in postmenopausal samples. Specifically, 390 lncRNAs were upregulated and 351 were downregulated by at least 1.5-fold in postmenopausal compared to premenopausal samples. Hierarchical clustering and TreeView analysis distinctly separated these lncRNAs by menopausal status (Figure 2A). The comparison between premenopausal and postmenopausal RNA-seq data is illustrated in the pie chart (Figure 2B). Furthermore, we identified 62 lncRNAs that exhibited greater than 1.5-fold change in expression exclusively in the postmenopausal group (Figure 2B,C). Gene ontology (GO) and KEGG pathway enrichment analysis of these 62 lncRNAs indicated predominant roles in the regulation of the actin cytoskeleton, focal adhesion, and the mTOR and FoxO signaling pathways (Figure 2D).

### 2.2. Validation of lncRNA Expression by qRT-PCR

We then validated the expression of nine differentially expressed lncRNAs (*LINC02433*, *LINC01449*, *ZEB2-AS1*, *SNHG12*, *LINC01117*, *CASC15*, *MIAT*, *H19*, and *HOTTIP*), identified from both our current and previous NGS analyses (Figure 2B). Validation by qRT-PCR was performed for these lncRNAs in an expanded cohort of postmenopausal (*n* = 31) and premenopausal (*n* = 84) paired samples. Among these, the expression of *LINC02433*, *LINC01449*, *ZEB2-AS1*, *SNHG12*, *CASC15*, *MIAT*, *H19* and *HOTTIP* have been previously reported [3,13,15,16,20,21], while *LINC01117* is novel. Figure 3 shows the fold-change expression levels (fib/paired myometrium) of nine lncRNAs, highlighting significant differences between the pre- and postmenopausal groups. The mean fold-changes oof *LINC02433*, *LINC01449*, *SNHG12*, *LINC01117*, *CASC15*, *MIAT*, *H19*, and *HOTTIP* were significantly lower in the postmenopausal group compared to the premenopausal group, while *ZEB2-AS1* was significantly upregulated.

Figure 4 presents the expression profiles of these nine lncRNAs in fib and myo tissues in pre- and postmenopausal groups. Several lncRNAs (*SNHG12*, *LINC01449*, and *LINC02433*, *H19*, and *HOTTIP*) were significantly upregulated in fib compared to myo in premenopausal samples, but this pattern was not observed in postmenopausal samples (Figure 4A,C). *CASC15* and *MIAT* were upregulated in fib relative to matched myo in both groups, but the magnitude of upregulation was reduced in postmenopausal samples (*CASC15*: 2.42-fold vs. 1.76-fold; *MIAT*: 3.61-fold vs. 1.84-fold) (Figure 4B,C). Interestingly, *ZEB2-AS1* was downregulated in fib compared to myo in premenopausal women but showed no significant change in postmenopausal samples. In contrast, *LINC01117* expression in premenopausal samples was not significantly altered, but this was significantly downregulated in fib from postmenopausal women (Figure 4B). Additionally, in Myo tissues, *LINC02433* expression was significantly higher in the postmenopausal group compared to the premenopausal group (Figure 4A), while *SNHG12* and *LINC01117* showed significantly lower expression in the postmenopausal group (Figure 4A,B). In Fib tissues, *ZEB2-AS1* was significantly upregulated in the postmenopausal group, whereas *LINC02433*, *LINC01449*, *SNHG12*, *MIAT*, *H19*, and *LINC01117* were significantly downregulated compared to the premenopausal group (Figure 4B).

We further analyzed the expression of these nine lncRNAs in relation to MED12 mutation status in both pre- and postmenopausal groups (Figure 5). *LINC01449* and *CASC15* were significantly upregulated in MED12–mutant samples compared to wild-type in both pre- and postmenopausal groups (Figure 5A,B). *MIAT* expression was also significantly higher in MED12–mutant samples in the premenopausal group but not in the postmenopausal group (Figure 5C). In the MED12 wild-type group, *ZEB2-AS1* expression was significantly higher in postmenopausal samples compared to premenopausal samples (Figure 5B), while *LINC02433*, *LINC01449*, *SNHG12*, and *LINC01117* were significantly downregulated in the postmenopausal group (Figure 5A,B). In the MED12–mutant group, *ZEB2-AS1* remained significantly upregulated in postmenopausal samples, whereas *LINC02433*, *LINC01449*, *CASC15*, *MIAT*, *H19*, and *LINC01117* were significantly downregulated compared to the premenopausal group (Figure 5).

A summary of findings from Figure 3, Figure 4 and Figure 5 is provided in Appendix A. Collectively, these data suggest that the expression of *CASC15*, *MIAT*, *H19*, *HOTTIP*, *ZEB2-AS1*, *LINC001117*, *SNHG12*, *LINC01449*, and *LINC02433* may be modulated by sex hormone levels.

## 3. Discussion

Using NGS, we identified 3684 differentially expressed lncRNAs, including 1702 upregulated and 1982 downregulated transcripts in Fib compared to Myo in postmenopausal uterine fibroids. Comparative analysis with our previously published premenopausal dataset revealed the expression of 741 differentially expressed lncRNAs in Fib was influenced by the menopausal status, including 62 lncRNAs which were uniquely dysregulated in postmenopausal samples. Pathway analysis indicated that these 62 lncRNAs were involved in pathways such as actin cytoskeleton regulation, focal adhesion, and mTOR/FoxO signaling. The validation of nine selected lncRNAs by PCR analysis, including *CASC15*, *MIAT*, *H19*, *HOTTIP*, *ZEB2-AS1*, *LINC001117*, *SNHG12*, *LINC01449*, and *LINC02433* in an expanded cohort confirmed distinct expression patterns for these lncRNAs based on the menopausal status and MED12 mutation status. These findings suggest that lncRNA expression in uterine fibroids is modulated by the menopausal status reflecting the potential influence of ovarian by sex hormones.

The role of estrogen and progesterone and their critical role in stimulating fibroid growth and progression are well established. Fibroids are characterized by overexpression of ER alpha and beta [4,22]. Furthermore, by virtue of higher expression of aromatase [23,24] and 17 βHSD [23] in the tumors as compared with myometrium local production of estradiol in fibroids is higher [25]. Estrogen also induces greater responsiveness to progesterone [26]. Progesterone also promotes fibroid growth [5,26,27] and fibroids are known to overexpress progesterone receptors [28,29]. Despite the abundant data on the effects of ovarian hormone on protein coding genes in fibroids there is scant data on their role in the regulation of ncRNAs. In silico analysis predicted estrogen receptor binding in 232 promoters of miRNA coding genes [30], and several studies have examined the effect of estrogen on miRNA expression in hormone responsive breast cancer cells [31,32]. The effect of estrogen on lncRNAs in ER positive breast cancer cells was also reported [33]. In the latter study [33] the transcription of estrogen regulated lncRNAs correlated with the activation status of ER alpha enhancers, chromatin accessibility and occupancy of enhancer regulators, P300, MED1 and ARID1B. Comparison of the lncRNA expression profile in premenopausal fibroids (GSE224991) with estrogen-regulated lncRNAs identified in breast cancer [33] revealed several overlapping transcripts, including *MIR503HG*, *LINC00511*, *RFPL1S*, and *CTB-178M22.2*, which were upregulated in both datasets. Interestingly, the expression of these lncRNAs was found to be reduced in postmenopausal fibroids in the current study. One of the limitations of this study is that circulating estrogen/progesterone levels were not available to directly correlate lncRNA expression levels with sex hormone levels. However, analysis using the ChIP-Atlas database [34], with a significance threshold set at 100 and including all available cell types, revealed that ESR1 potentially regulates all nine lncRNAs. PGR was found to regulate *CASC15*, *MIAT*, *HOTTIP*, *LINC001117*, *SNHG12*, *LINC01449*, and *LINC02433*. These findings support the potential influence of sex hormones on fibroid-associated lncRNA expression.

In terms of the role of progesterone in the regulation of lncRNA transcription there are currently no available studies. Notable exceptions include our in vitro study demonstrating the stimulatory effect of progesterone on *XIST* expression [11] and the in vitro study by Cao et al. where the combination of estrogen and progesterone but not progesterone alone significantly stimulated *H19* mRNA levels in fibroid cells [16]. The presence of ERE and PRE in the differentially expressed lncRNAs we have identified in postmenopausal fibroids is supportive of their hormone dependence; however other mechanisms independent of estrogen and progesterone effects could also account for this differential pattern of lncRNA expression in fibroids from postmenopausal women. Several differentially expressed lncRNA in fibroid from postmenopausal women have been shown to have oncogenic roles in different types of cancer including *HOTTIP* [35,36,37], *SNHG12* [38,39], *CASC15* [40,41], *ZEB2-AS1* [42]. The Wnt-beta catenin pathway, a well-established critical tumorigenic pathway in fibroid [43] also mediates the action of some of these oncogenic lncRNAs in a tissue dependent manner [36,41,44,45,46]. However, the regulatory role of ovarian steroids on these lncRNAs remains to be determined. *CASC15* is a super enhancer lncRNA which we previously reported to be overexpressed in fibroids along with its corresponding coding transcript, prolactin [20]. Prolactin is one of the most highly differentially overexpressed genes in fibroids shown to be secreted by fibroid cell [47,48]. Prolactin stimulates fibroid cell proliferation [49], and contributes to ECM accumulation characteristic of fibroid through its effect promoting myofibroblast differentiation, an effect mediated by MAPK and STAT5 signaling [50]. Both estrogen and progesterone are known to stimulate the expression of prolactin and its receptor in the uterus [51]. The functional role and influence of ovarian steroids on other differentially expressed lncRNAs identified here including *LINC01449*, *LINC02433* and *LINC01117* remain unknown.

In our previous study MED12 mutation had a significant effect on the expression of a number of lncRNAs in Fib from premenopausal women [3]. In that study MED12 mutation in the tumor was found to increase the magnitude of differential expression of a number of lncRNAs (*TPTEP1*, *PART1*, *RPS10P7*, *MSC-AS1*, *LINC00337*, *LINC00536*, *LINC01436*, *LINC01449*, *LINC02433*, *LINC01186*, and *LINC02624*). The current study also shows MED12 mutation has an effect on lncRNA expression in Fib from postmenopausal women however this effect is blunted, suggesting that the mutation effect on lncRNA expression diminishes over time. Due to the limited sample size, the impact of race/ethnicity on lncRNA expression in postmenopausal samples could not be determined.

In summary, we have identified a number of differentially expressed lncRNAs in fibroids from postmenopausal women in whom estrogen and progesterone production is significantly diminished. These lncRNAs show a different pattern of expression than fibroids from premenopausal women suggesting that their expression is hormonally linked. Several of these lncRNAs are known to have oncogenic roles in various types of cancer. Further studies will be needed to establish the role of estrogen and progesterone on lncRNA expression in fibroids.

## 4. Materials and Methods

### 4.1. Myometrium and Fibroid Tissues Collection

To minimize variability among Fib samples, tumors measuring 3 to 5 cm in diameter and located intramurally were collected from premenopausal (*n* = 84; mean age 44 ± 4.2 years) and postmenopausal (*n* = 31; mean age 58 ± 5.0 years) women undergoing hysterectomy for symptomatic fibroids at Harbor-UCLA Medical Center. The racial distribution in premenopausal specimens was as follows: White (*n* = 10), Hispanics (*n* = 50), Black (22), and Asian (2), and in postmenopausal specimens was White (*n* = 2); Hispanics (*n* = 20), Black (*n* = 8) and Asian (*n* = 1). The study was approved by the Institutional Review Board at the Lundquist Institute (IRB# 18CR-31752-01R), and written informed consent was obtained from all participants. None of the subjects had received hormonal therapy for at least three months prior to surgery. Tissue specimens were snap-frozen and stored in liquid nitrogen for further analysis, as previously described [52,53].

### 4.2. MED12 Mutation Analysis

Genomic DNA was extracted from 100 mg of freshly frozen fibroid and matched myometrial tissues from both premenopausal and postmenopausal women using the MagaZorb DNA Mini-Prep Kit (Promega, Madison, WI, USA), following the manufacturer’s instructions. MED12 exon 2 mutations were analyzed by PCR amplification and Sanger sequencing (Transnetyx Inc., Culver City, CA, USA) using previously reported primers: sense 5′-GCCCTTTCACCTTGTTCCTT-3′ and antisense 5′-TGTCCCTATAAGTCTTCCCAACC-3′ [13]. In the premenopausal group (*n* = 84), MED12 mutations were detected in 60 Fib (71.4%), with no mutations found in the corresponding myometrial tissues. The majority were missense mutations in exon 2 (52/60), with the remainder being in-frame insertion–deletion variants (8/60). The missense mutations included: c.130G>C (p.Gly44Arg, 8 cases), c.130G>A (p.Gly44Ser, 8), c.130G>T (p.Gly44Cys, 4), c.131G>C (p.Gly44Ala, 3), c.131G>A (p.Gly44Asp, 18), c.131G>T (p.Gly44Val, 8), c.128A>C (p.Gln43Pro, 1), and c.107T>G (p.Leu36Arg, 2). In the postmenopausal group (*n* = 31), MED12 mutations were identified in 19 Fib (61.3%), with no mutations observed in the paired myometrium. Fifteen of these were missense mutations and four were in-frame insertion–deletion types. The missense variants included c.130G>C (p.Gly44Arg, 1), c.130G>A (p.Gly44Ser, 3), c.131G>C (p.Gly44Ala, 1), and c.131G>A (p.Gly44Asp, 10).

### 4.3. RNA Sequencing and Bioinformatic Analysis

Total RNA was extracted from six paired postmenopausal fibroid and myometrial tissues using TRIzol reagent (Thermo Fisher Scientific, Waltham, MA, USA). RNA concentration and integrity were assessed with a NanoDrop 2000c spectrophotometer (Thermo Scientific, Wilmington, DE, USA) and an Agilent 2100 Bioanalyzer (Agilent Technologies, Santa Clara, CA, USA), as previously described [54,55]. Only samples with RNA integrity numbers (RIN) ≥9 were used for library preparation. One microgram of total RNA per sample was used to generate strand-specific cDNA libraries using the TruSeq protocol (Illumina, San Diego, CA, USA) following the manufacturer’s instructions. RNA sequencing was performed at the UCLA Technology Center for Genomics & Bioinformatics. Bioinformatics analysis was conducted as previously described [24,29]. For quality control FastQC was used to check the quality of raw fastq data from sequencing core and after adaptor cut and quality trimming [56]. The sequencing reads were mapped by STAR 2.7.9a [57] and read counts per gene were quantified using the human genome GRCh38. RNA-seq count data were normalized using the DESeq2 R package (version 1.38.3) [58]. Differential gene expression was visualized using hierarchical clustering and TreeView analysis, volcano plots, and principal component analysis (PCA) using Flaski (version 3.19.3) [59]. Functional enrichment analyses, including gene ontology (GO) and KEGG pathway analysis, were performed using NcPath, which integrates data from org.Hs.eg.db (release 3.11), miRBase v22.1, miRTarBase v8, RNAinter, and NPInter v4.0 [60]. The observed expression differences were deemed robust and suitable for downstream statistical analysis. The RNA sequencing dataset is publicly available in the Gene Expression Omnibus (GEO) under accession number GSE302523.

### 4.4. Quantitative RT-PCR

Briefly, 2 μg of total RNA was reverse transcribed into cDNA using random primers following the manufacturer’s instructions (Applied Biosystems, Carlsbad, CA, USA). Quantitative real-time PCR (qRT-PCR) was performed using SYBR Green Gene Expression Master Mix (Applied Biosystems) as previously described [61]. Gene expression levels were quantified on the Invitrogen StepOne Real-Time PCR System, with FBXW2 (F-box and WD repeat domain-containing 2) used as the endogenous control for normalization [62]. All reactions were conducted in triplicate. Relative mRNA expression levels were calculated using the comparative cycle threshold (2^−ΔΔCq^) method according to the manufacturer’s protocol (Applied Biosystems). Results were expressed as fold change relative to the control group. Primer sequences used for target genes are listed in Appendix A.

### 4.5. Statistical Analysis

All data are presented as mean ± standard error of the mean (SEM) and were analyzed using GraphPad Prism software 10.5.0 (GraphPad, San Diego, CA, USA). Normality of the datasets was assessed using the Kolmogorov–Smirnov test. As the data did not follow a normal distribution, non-parametric Mann–Whitney tests were applied for statistical comparisons. A *p*-value of <0.05 was considered statistically significant.

## Figures and Tables

**Figure 1 ijms-26-06798-f001:**
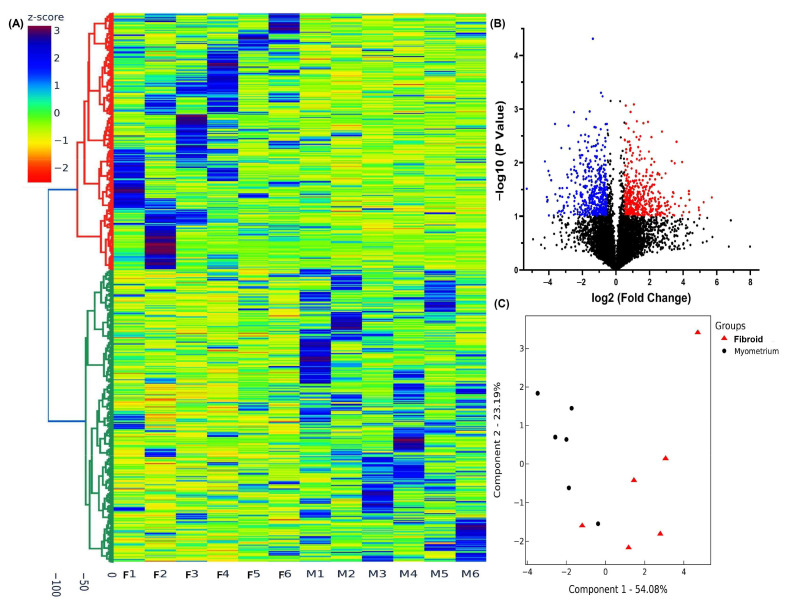
Heterogeneity and transcriptomic changes in fibroids compared to myometrium. (**A**) Hierarchical clustered heatmap analysis of differentially expressed lncRNAs (fold change ≥1.5, *p* < 0.05) in 6 paired postmenopausal fibroids and matched myometrium. Color gradient represents gene expression as z-scores. (**B**) Volcano plot showing significantly upregulated (red; *n* = 461) and downregulated genes (blue; *n* = 497) with adjusted *p* < 0.05. (**C**) Principal component analysis (PCA) plot of RNA-seq results from paired fibroid and matched myometrium (*n* = 6). Each dot represents one sample. Myometrial samples (Myo) are shown in black and fibroid samples (Lyo) are shown in red.

**Figure 2 ijms-26-06798-f002:**
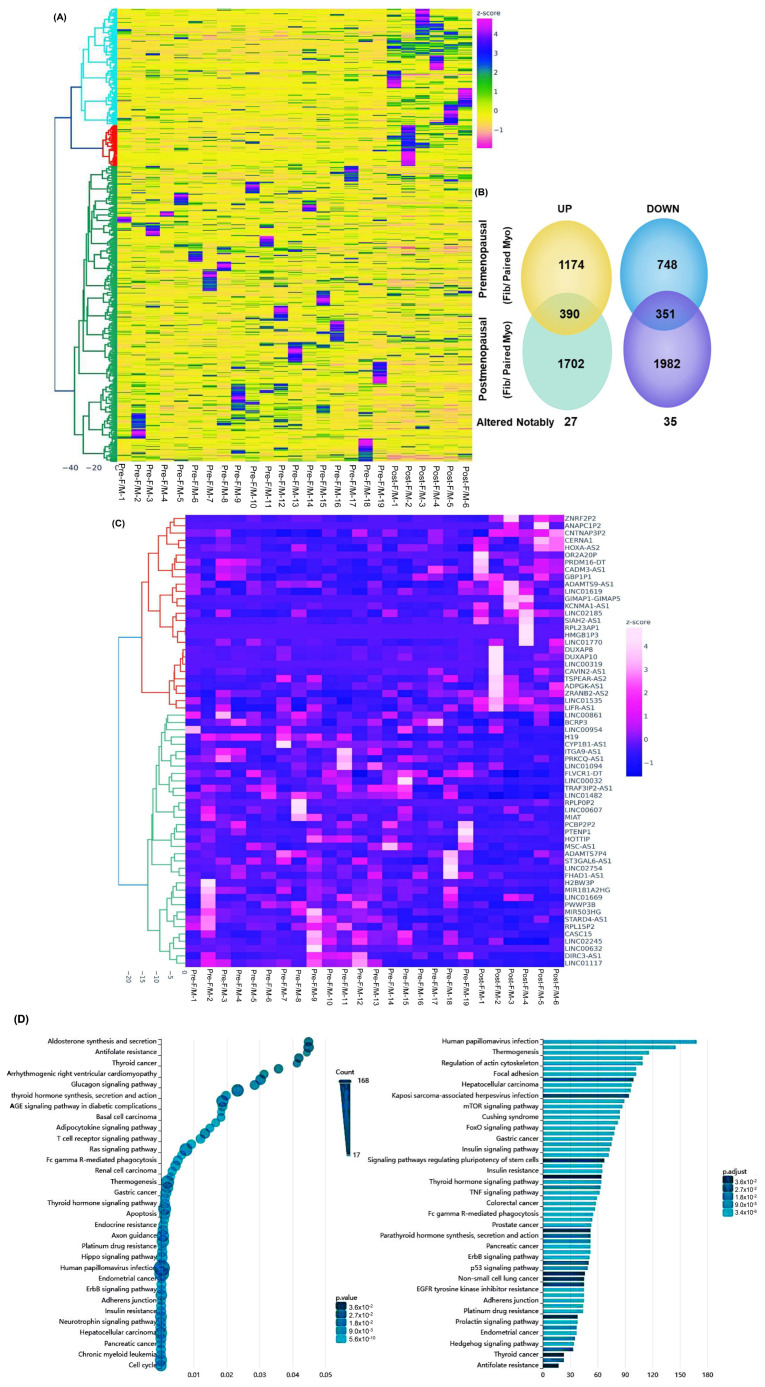
Differential levels of transcriptomic heterogeneity of lncRNAs in paired specimens analyzed based on menopausal status. (**A**) Hierarchical clustered heatmap analysis was performed as fold change (fibroid/paired myometrium), comparing premenopausal (*n* = 19) with postmenopausal group (*n* = 6) (fold change ≥1.5, *p* < 0.05). Color gradient represents gene expression as z-scores. (**B**) Pie chart illustrates comparison between premenopausal and postmenopausal RNA-seq data. Central circle highlights 62 lncRNAs that were significantly altered in postmenopausal group, including 27 upregulated and 35 downregulated transcripts. (**C**) Heatmap of 62 lncRNAs (fibroid/paired myometrium) differentially expressed exclusively in the postmenopausal group (fold change ≥1.5, *p* < 0.05). Color gradient represents gene expression levels as z-scores. (**D**) Gene ontology (GO) analysis of 62 differentially expressed lncRNAs. Color gradient represents levels of log2 fold change presented as z-scores.

**Figure 3 ijms-26-06798-f003:**
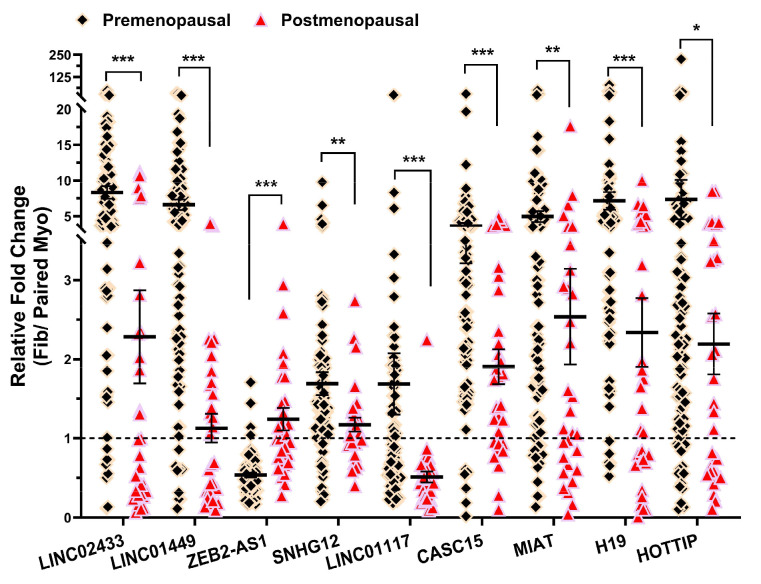
Expression of lncRNAs for SNHG12, LINC02433, LINC01449; ZEB2-AS1, LINC01117, and CASC15; H19, HOTTIP, and MIAT expressed as fold change (fib/paired myo) in premenopausal (*n* = 84) and postmenopausal group (*n* = 31) by qRT-PCR. The results are presented as mean ± SEM with *p* values (* *p* < 0.05; ** *p* < 0.01; *** *p* < 0.001) as indicated by the corresponding lines.

**Figure 4 ijms-26-06798-f004:**
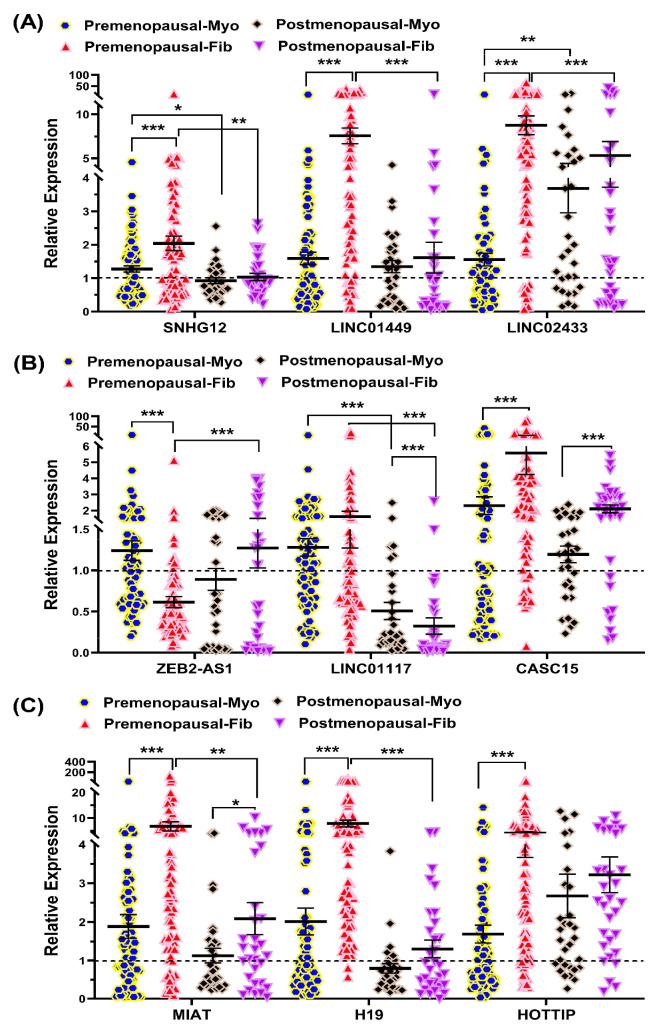
Expression of select lncRNAs in myometrium (myo) and paired fibroids (fib) in premenopausal (*n* = 84) and postmenopausal group (*n* = 31) as assessed by qRT-PCR. (**A**) *SNHG12*, *LINC02433*, *LINC01449*; (**B**) *ZEB2-AS1*, *LINC01117*, *CASC15*; (**C**) *H19*, *HOTTIP*, and *MIAT*. Results are presented as mean ± SEM with *p* values (* *p* < 0.05; ** *p* < 0.01; *** *p* < 0.001), indicated by corresponding lines.

**Figure 5 ijms-26-06798-f005:**
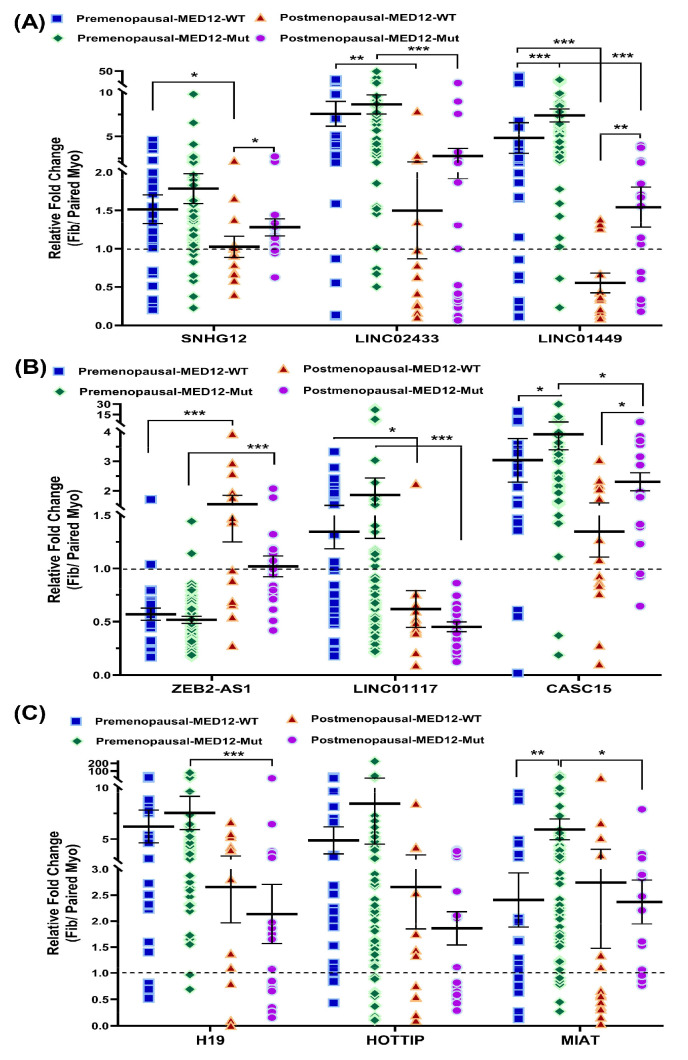
The expression of lncRNA for (**A**) *SNHG12*, *LINC02433*, *LINC01449*; (**B**) *ZEB2-AS1*, *LINC01117*, *CASC15*; (**C**) *H19*, *HOTTIP*, and *MIAT* in premenopausal MED12 mutation-negative samples (WT, *n* = 24) and mutation-positive samples (Mut, *n* = 60) and postmenopausal MED12 mutation-negative samples (*n* = 12) and mutation-positive samples (*n* = 19) as assessed by qRT-PCR. Results are presented as mean ± SEM with *p* values (* *p* < 0.05; ** *p* < 0.01; *** *p* < 0.001) as indicated by corresponding lines.

## Data Availability

Raw data were generated at The Lundquist Institute. Derived data supporting the findings of this study are available from the corresponding author O.K. on request.

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
