# Peer review of "Comparative Analysis of Differentially Expressed Long Non-Coding RNA in Pre- and Postmenopausal Fibroids"

_ijms, 2025, doi:10.3390/ijms26146798_

Round 1

Reviewer 1 Report

Comments and Suggestions for Authors

The paper is well written. It is interesting to reader to undserstand the profiles of differentially  expressed long non-coding RNAs (lncRNAs) in fibroids from postmenopausal and premenopausal women to identify hormone-responsive lncRNAs. I think the paper is suitable for publishing in IJMS.

Minor revision should be concerned.

  1. L52 delete “and can be” and add“,”
  2.  L60 replace "some" as "certain "
  3. L68  replace "Fib " as "fibroids (Fib)", replace "Myo " as "myometrium (Myo)",
  4. L70  replace "the Hierarchical“ as ”hierarchical“
  5. L77  replace "to“ as ”with“
  6. Figure 2 is not sufficiently clear. Please replace it with a higher-resolution or more detailed version.
  7. L209-210 rewrote  the sentence. Please see L209-210. Notable exceptions include our in vitro study demonstrating the stimulatory effect of progesterone on XIST expression.
  8. L210 delete "of the"
  9. delete "which is" replace "which is" as" a"
  10. The sentence "the role of 222 ovarian steroids in their regulation remains undetermined" corrected as "the regulatory role of ovarian steroids on these lncRNAs remains to be determined"
  11. delete "The effect of race/ethnicity" and corretted as "Due to the limited sample size, the impact of race/ethnicity"; delete "due to limited sample size".
  12. L255 add "as follows: "
  13. L281?Why did the authors select six pairs of fibroid (Fib) and adjacent myometrium (Myo) tissues? PleLase explain it. I think it is limited.
  14.  L291 What are housekeeping genes? Please provide additional information.
  15.  

Author Response

The paper is well written. It is interesting to reader to undserstand the profiles of differentially expressed long non-coding RNAs (lncRNAs) in fibroids from postmenopausal and premenopausal women to identify hormone-responsive lncRNAs. I think the paper is suitable for publishing in IJMS.

Minor revision should be concerned.

L52 delete “and can be” and add“,”

 L60 replace "some" as "certain "

L68  replace "Fib " as "fibroids (Fib)", replace "Myo " as "myometrium (Myo)",

L70  replace "the Hierarchical“ as ”hierarchical“

L77  replace "to“ as ”with“

Figure 2 is not sufficiently clear. Please replace it with a higher-resolution or more detailed version.

L209-210 rewrote  the sentence. Please see L209-210. Notable exceptions include our in vitro study demonstrating the stimulatory effect of progesterone on XIST expression.

L210 delete "of the"

delete "which is" replace "which is" as" a"

The sentence "the role of 222 ovarian steroids in their regulation remains undetermined" corrected as "the regulatory role of ovarian steroids on these lncRNAs remains to be determined"

delete "The effect of race/ethnicity" and corretted as "Due to the limited sample size, the impact of race/ethnicity"; delete "due to limited sample size".

L255 add "as follows: "

Response: Thanks for your suggestions. We edited our manuscript as suggested.

L281?Why did the authors select six pairs of fibroid (Fib) and adjacent myometrium (Myo) tissues? PleLase explain it. I think it is limited.

Response: We selected six pairs of fibroid and matched myometrium from postmenopausal women for RNAseq. However, we perform validation by qRT-PCR using 84 paired premenopausal and 31 paired postmenopausal specimens. We believe it is not limited.

 L291 What are housekeeping genes? Please provide additional information.

Response: We added one sentence in L292 as below:

“RNA-seq count data were normalized using the DESeq2 R package (version 1.38.3)”

Reviewer 2 Report

Comments and Suggestions for Authors

General Comments:

This is a well-executed and timely study exploring lncRNA expression profiles in fibroids from postmenopausal versus premenopausal women, an important but underexplored area. The integration of RNA-seq data with previously published datasets and the validation in an expanded cohort are strong aspects of the work. The findings are novel and provide meaningful insight into hormone-responsive regulatory mechanisms in fibroid biology. Minor revisions aimed at clarifying methodology, strengthening data interpretation, and improving contextual framing will further enhance the manuscript’s impact.

Minor comments:

  • If possible, please include information on the hormonal status (e.g., estradiol/progesterone levels) of the pre- and postmenopausal patients. This would add clinical depth and increase the impact of the study.

  • The paper would have benefited from including a comparison between myometrium samples from pre- and postmenopausal women, as well as between fibroids from these two groups. Such comparisons would have provided a more comprehensive view of menopause-associated transcriptomic changes.

  • Ensure consistency in terminology for sample types throughout the text and figures. For example, “Fib” and “Myo” are used in the text, while “L” and “M” appear in Figure 1. Please standardize this across all sections.

  • Line 97: The sentence identifies HOTTIP as a novel lncRNA in fibroids, but this contradicts prior literature. HOTTIP has been previously reported in fibroids (PMID: 31851934) and should not be labeled as novel.

  • Please make p-value/FDR annotations consistent throughout the manuscript. For example, line 73 usesp < 0.05, while the Figure 1 legend uses FDR < 0.05. Choose one and apply it uniformly.

  • In the description of Figure 4 (“the magnitude of increase was less pronounced in postmenopausal samples”), please report actual fold changes to support the interpretation.

  • Similarly, in the description of Figure 5 (“although the increase was less pronounced in the postmenopausal group”), please quantify the differences using fold changes.

  • Figure 2B: Consider replacing the current format with a Venn diagram to make the overlap and exclusivity of dysregulated lncRNAs across groups easier to understand.

  • Figure 2 caption: Add the termdifferentially to the description, “Heatmap of the 62 lncRNAs differentially expressed exclusively...”

  • Please include key details of the RNA-seq pipeline, such as the read trimming tool, genome version used for alignment, and the software/packages (with version numbers) used for differential expression analysis and generating the heatmap, PCA, and dot plots. 

Author Response

This is a well-executed and timely study exploring lncRNA expression profiles in fibroids from postmenopausal versus premenopausal women, an important but underexplored area. The integration of RNA-seq data with previously published datasets and the validation in an expanded cohort are strong aspects of the work. The findings are novel and provide meaningful insight into hormone-responsive regulatory mechanisms in fibroid biology. Minor revisions aimed at clarifying methodology, strengthening data interpretation, and improving contextual framing will further enhance the manuscript’s impact.

Minor comments:

If possible, please include information on the hormonal status (e.g., estradiol/progesterone levels) of the pre- and postmenopausal patients. This would add clinical depth and increase the impact of the study.

 Response: Thanks for your suggestion. Unfortunately, we don’t have the information of the hormonal status of the pre- and postmenopausal patients. We have included this limitation of the study in the Discussion section:

“One of the limitations of this study is that circulating estrogen/progesterone levels were not available to directly correlate lncRNA expression levels with sex hormone levels.”

The paper would have benefited from including a comparison between myometrium samples from pre- and postmenopausal women, as well as between fibroids from these two groups. Such comparisons would have provided a more comprehensive view of menopause-associated transcriptomic changes.

 Response: Thanks for your suggestion. We included the information in the results section:

“Additionally, the relative expression level of LINC02433 in Myo was significantly higher in the postmenopausal group compared to the premenopausal group (Fig. 4A). In contrast, SNHG12 and LINC01117 showed significantly lower expression in Myo of the postmenopausal group (Fig. 4A and 4B). In Fib tissues, ZEB2-AS1 expression was significantly higher in the postmenopausal group, whereas LINC02433, LINC01449, SNHG12, MIAT, H19, and LINC01117 were significantly downregulated compared to the premenopausal group (Fig. 4).”

Ensure consistency in terminology for sample types throughout the text and figures. For example, “Fib” and “Myo” are used in the text, while “L” and “M” appear in Figure 1. Please standardize this across all sections.

Response: Thanks for your suggestion. We updated all the figures to have consistency with the text.

Line 97: The sentence identifies HOTTIP as a novel lncRNA in fibroids, but this contradicts prior literature. HOTTIP has been previously reported in fibroids (PMID: 31851934) and should not be labeled as novel.

Response: Thanks for your suggestion. We edited the sentence and added the suggested reference.

Please make p-value/FDR annotations consistent throughout the manuscript. For example, line 73 usesp < 0.05, while the Figure 1 legend uses FDR < 0.05. Choose one and apply it uniformly.

Response: Thanks for your suggestion. We edited the figure legend of Fig. 1B as:

"Volcano plot showing significantly upregulated (red; n = 461) and downregulated genes (blue; n = 497) with adjusted p < 0.05."

In the description of Figure 4 (“the magnitude of increase was less pronounced in postmenopausal samples”), please report actual fold changes to support the interpretation.

Response: Thanks for your suggestion. We added the actual fold change information in the text:

“CASC15 and MIAT exhibited higher expression in Fib samples compared to matched Myo in both groups; however, the magnitude of upregulation was less pronounced in postmenopausal samples (premenopausal vs. postmenopausal, CASC15: 2.42-fold vs. 1.76-fold; MIAT: 3.61-fold vs. 1.84-fold) (Figs. 4B and 4C).

Similarly, in the description of Figure 5 (“although the increase was less pronounced in the postmenopausal group”), please quantify the differences using fold changes.

Response: Thanks for your suggestion. We realized we had misinterpreted our results and have revised them as follows:

“We further analyzed the expression of these nine lncRNAs in relation to MED12 mutation status in both pre- and postmenopausal groups (Fig. 5). LINC01449 and CASC15 were significantly upregulated in MED12-mutant samples compared to wild-type in both pre- and postmenopausal groups (Figs. 5A and 5B). MIAT expression was also significantly higher in MED12-mutant samples in the premenopausal group but not in the postmenopausal group (Fig. 5C). In the MED12 wild-type group, ZEB2-AS1 expression was significantly higher in postmenopausal samples compared to premenopausal samples (Fig. 5B), while LINC02433, LINC01449, SNHG12, and LINC01117 were significantly downregulated in the postmenopausal group (Figs. 5A and 5B). In the MED12-mutant group, ZEB2-AS1 remained significantly upregulated in postmenopausal samples, whereas LINC02433, LINC01449, CASC15, MIAT, H19, and LINC01117 were significantly downregulated compared to the premenopausal group (Fig. 5).”

Figure 2B: Consider replacing the current format with a Venn diagram to make the overlap and exclusivity of dysregulated lncRNAs across groups easier to understand.

Response: Thanks for your suggestion. We replaced with Venn diagram format.

Figure 2 caption: Add the term“differentially” to the description, “Heatmap of the 62 lncRNAs differentially expressed exclusively...”

Response: We added as suggested.

Please include key details of the RNA-seq pipeline, such as the read trimming tool, genome version used for alignment, and the software/packages (with version numbers) used for differential expression analysis and generating the heatmap, PCA, and dot plots.

 Response: We updated all requested information in the revision.

“For quality control FastQC was used to check the quality of raw fastq data from se-quencing core and after adaptor cut and quality trimming [56]. The sequencing reads were mapped by STAR 2.7.9a [57] and read counts per gene were quantified using the human genome GRCh38. RNA-seq count data were normalized using the DESeq2 R package (version 1.38.3) [58]. Differential gene expression was visualized using hier-archical clustering and TreeView analysis, volcano plots, and principal component analysis (PCA) using Flaski (version 3.19.3) [59]. Functional enrichment analyses, including Gene Ontology (GO) and KEGG pathway analysis, were performed using NcPath, which integrates data from org.Hs.eg.db (release 3.11), miRBase v22.1, miRTarBase v8, RNAinter, and NPInter v4.0. [60].”